# Enhanced Predictive Value of Lipid Accumulation Product for Identifying Metabolic Syndrome in the General Population of China

**DOI:** 10.3390/nu15143168

**Published:** 2023-07-17

**Authors:** Qi Shao, Jing Li, Yiling Wu, Xing Liu, Na Wang, Yonggen Jiang, Qi Zhao, Genming Zhao

**Affiliations:** 1Key Laboratory of Public Health Safety of Ministry of Education, Department of Epidemiology, School of Public Health, Fudan University, Shanghai 200032, China; shaoqi0045@126.com (Q.S.); liuxing@fudan.edu.cn (X.L.); na.wang@fudan.edu.cn (N.W.); gmzhao@shmu.edu.cn (G.Z.); 2Zhongshan Community Health Center, Shanghai 201613, China; zhongshanlijing@163.com; 3Songjiang District Center for Disease Control and Prevention, Shanghai 201600, China; aries2119@163.com (Y.W.); sjjkzx1106@126.com (Y.J.); 4NHC Key Laboratory of Health Technology Assessment, Department of Social Medicine, School of Public Health, Fudan University, Shanghai 200032, China

**Keywords:** lipid accumulation product, metabolic syndrome, obesity indicators

## Abstract

The purpose of this research was to evaluate the lipid accumulation product (LAP)’s accuracy and predictive value for identifying metabolic syndrome (MS) in the general Chinese population compared with other obesity indicators. Baseline survey information from a population-based cohort study carried out in Shanghai’s Songjiang District was used in this research. Odds ratios (OR) and a 95% confidence interval (CI) were obtained by logistic regression. The ability of each variable to detect MS was assessed using the receiver operating characteristic curve (ROC). The optimum cut-off point for each indicator was selected using Youden’s index. The survey involved 35,446 participants in total. In both genders, the prevalence of MS rose as the LAP increased (*p* < 0.001). The LAP’s AUC was 0.901 (95%CI: 0.895–0.906) in males and 0.898 (95%CI: 0.893–0.902) in females, making it substantially more predictive of MS than other variables (BMI, WC, WHR, WHtR). The optimal cutoff point of the LAP for men and women was 36.04 (Se: 81.91%, Sp: 81.06%) and 34.95 (Se: 80.93%, Sp: 83.04%). The Youden index of the LAP was 0.64 for both sexes. Our findings imply that the LAP, compared to other obesity markers in China, is a more accurate predictor of MS.

## 1. Introduction

Metabolic syndrome is a typical metabolic disorder. This condition is brought on by the increasing prevalence of obesity. A cluster of metabolic and cardiovascular risk factors known as metabolic syndrome includes central obesity, dyslipidemia, hypertension, hyperinsulinemia, and insulin resistance (IR) [1,2]. Mellitus type II, cardiovascular diseases (CVD), malignancies, and overall mortality are all strongly correlated with MS [3,4,5,6,7]. Additionally, there is a strong link between MS and psychological strain [8], as well as mental disorders, including depression, schizophrenia, and bipolar disorder [9,10]. The prevalence of metabolic syndrome is gradually rising in China, despite the fact that it is lower than that of European and American nations (approximately 30% [11,12]), as a result of the country’s enormous changes, such as the population’s aging trend, declining birthrate, and economic transformation [13,14]. Early diagnosis and timely health measures can effectively prevent the aggravation of this and other diseases. Therefore, early identification and interventions of metabolic syndrome have become a hot topic in order to reduce the occurrence and development of related serious conditions, and it is quite necessary to find a simple and workable indicator for the diagnosis of people at risk for MS in clinical settings, particularly in screening assessments.

Waist circumference (WC) and fasting triglycerides (TG) are used to calculate the lipid accumulation product (LAP), which describes the excess lipid accumulation proposed by Kahn [15]. In subsequent studies, it was discovered that the LAP had superior diagnostic and predictive power to traditional indicators, such as BMI in a variety of diseases [16,17,18,19,20,21,22,23,24], such as diabetes, CVD, hyperuricemia, hypertension, and influenced all-cause mortality [17,25]. It is still unknown, nevertheless, whether the LAP can accurately predict MS in China today and whether the predictive power is strong.

Therefore, this study used the survey data of adult citizens living in Songjiang District of Shanghai, China to evaluate the correlation between the LAP and other obesity indicators and MS, to compare the predictive power of each indicator, and to find the best cut-off point, which can provide a reference for the judgment and screening of MS.

## 2. Materials and Methods

### 2.1. Study Design and Participants

This study was conducted based on baseline survey data collected from the Shanghai Suburban Adult Cohort and Biobank (SSACB) study [26], a community-based natural population cohort study conducted by the School of Public Health, Fudan University from April 2016 to October 2017. To select the study participants, a multistage, stratified, clustered sampling design was used. In the first phase, 4 communities were randomly chosen based on their location and economic standing: 2 urban communities (Zhongshan and Xinqiao), 1 urban-rural mixed area (Sheshan), and 1 rural area (Maogang). In the second phase, 16 administrative villages from Maogang and 9, 18, and 4 community committees from Xinqiao, Zhongshan, and Sheshan, respectively, were chosen at random. In the third phase, people aged between 20 and 74 who are natives of the city or have been in Shanghai for a minimum of five years were selected as research subjects [26]. Overall, a total of 36,403 residents signed up for the study. In total, 35,446 candidates were included in the investigation after 957 people who lacked crucial data were excluded. The Fudan University School of Public Health’s Ethics Committee provided approval for the study. (IRB number 2016-04-0586).

### 2.2. Data Collection

#### 2.2.1. Questionnaires

Following a physical examination at their neighborhood community health service center, each participant was asked to fill out a structured questionnaire. The questionnaire interviews were conducted using an Android tablet, which allowed for paperless data entry and audio recordings for later listening [26]. The questionnaire included sociodemographic characteristics (including age, gender, etc.), lifestyle habits (including smoking, drinking, exercise, etc.), and personal and family history of diseases (including personal disease, family history of disease, etc.) of the respondents. Smoking was deemed to have occurred if at least one cigarette was consumed every day for at least six months. Drinking was defined as consuming alcohol at least three times per week for at least six months. Exercise was considered as physical exercise that lasted at least 10 min each week [26].

#### 2.2.2. Anthropometric Measurements

A metal column height meter was used to measure height (exact to 0.1 cm), and an electronic weight scale was used to measure weight (exact to 0.1 kg). Body weight divided by the square of standing height was used to calculate the BMI. Following the reference standard of the Chinese body mass index, BMI was divided into four distinct groups: underweight (<18.5 kg/m^2^), normal (18.5 kg/m^2^–23.9 kg/m^2^), overweight (24 kg/m^2^–27.9 kg/m^2^), and obese (>28 kg/m^2^) [27].

A flexible ruler was used to measure the hip and waist circumferences. The hip circumference was obtained at the horizontal circumference of the most prominent section of the buttocks, and the waist circumference was observed at the midpoint of the line between the costal margin and the anterior superior iliac spine. The measurement was accurate to 0.1 cm when the flexible ruler was placed flat on the skin without compression. The ratio of waist circumference to hip circumference is the waist-to-hip ratio (WHR), and the waist-to-height ratio (WHtR) was calculated from the ratio of waist circumference to height.

After a 5-min break, blood pressure was checked utilizing a digital sphygmomanometer on the right arm in a quiet environment. Three readings were averaged to determine the mean (the blood pressure was accurate to 1 mmHg, 0.133 kPa).

#### 2.2.3. Biological Sample Collection and Determination

After a fast for an entire night, blood samples were taken the next morning. A 16 mL fasting blood sample was drawn into four vacutainers; 2 mL was drawn into a tube which contained EDTA and 4 mL was drawn using serum-separating tubes, both of which were sent to clinical laboratories (the Shanghai Dian Diagnostics Co. Ltd.) for testing; the other 10 mL was centrifuged, serum and blood clotting factors were aliquoted, and the sample was stored at 80 °C for future use/biobank [26]. The test result data used in this study came from the test of serum. The Roche COBASC501 automatic biochemical analyzer was utilized to quantify serum total cholesterol (TC), triglycerides (TG), low-density lipoprotein cholesterol (LDL-C), and high-density lipoprotein cholesterol (HDL-C). The glycosidase method was used to quantify the fasting plasma glucose (FPG) (Roche P800 automatic biochemical analyzer). High-pressure liquid chromatography (TOSOH G8, Automatic Hemoglobin A1c Analyzer) was used to measure glycated hemoglobin (HbA1c) [26].

### 2.3. Assessment of Lipid Accumulation Product

For men, the formula for calculating the LAP was [WC (cm)–65] × TG concentration (mmol/L), whereas, for women, it was [WC (cm)–58] × TG concentration (mmol/L) [15,28]. According to Kahn, any WC value of 65 cm or less in men was corrected upward to 66.0 cm (*n* = 209), and any values of 58 cm or less in women was updated upward to 59.0 cm (*n* = 105), in order to avoid obtaining non-positive values for the LAP [15].

### 2.4. Definition of MS

The joint interim statement (JIS) “Harmonized” criteria for MS was implemented in this investigation [29,30]. The presence of 3 or more of the following criteria led to the diagnosis of metabolic syndrome: (1) WC_male_ ≥ 90 cm and WC_female_ ≥ 80 cm; (2) TG ≥ 1.7 mmol/L; (3) HDL-c, <1.0 mmol/L for men and <1.3 mmol/L for women; (4) elevated blood pressure, defined as a systolic blood pressure (SBP) ≥ 30 mmHg or a diastolic blood pressure (DBP) ≥ 85 mmHg, or under current use of anti-hypertensive medications; and (5) hyperglycemia, defined as fasting blood sugar ≥ 5.6 mmol/L, or under current use of anti-diabetic medications.

### 2.5. Definition of the Relevant Indicators

Accuracy refers to the percentage of the correct number of samples in the total. Sensitivity is the percentage of actual illnesses correctly identified as sick by this diagnostic criterion. It reflects the ability of the screening tests to detect patients. Speciality is the percentage of actual disease-free being correctly judged as disease-free according to this diagnostic criterion. It reflects the ability of screening tests to identify excluded patients. The Youden index is the method for evaluating the authenticity of a screening test. It represents the total ability of screening methods to detect true patients versus non-patients. Positive predictive value (PPV) refers to the proportion of the number of truly “sick” cases of the total number of positive cases detected by the diagnostic criteria, reflecting the likelihood of the positive screening test results of having the target disease. The positive likelihood ratio (+LR) is the ratio of the true positive rate to the false positive rate in the screening result.

### 2.6. Statistical Analysis

Version 26 of the IBM SPSS Statistics was used to analyze the data. The Kolmogorov-Smirnov test was applied to evaluate whether data were regularly distributed. If the normal distribution was satisfied, the continuous variable is represented as mean ± standard deviation (SD), and if not, as the median and interquartile range (IQR). Differences were analyzed by using the analysis of variance or the Kruskal-Wallis H test for continuous variables. The categorical variables were displayed as frequency (n) and proportion (%), and the chi-square test was applied for comparison between groups. Using quartiles, the LAP was classified into four categories (Q1, Q2, Q3, Q4).

Using the odds ratio (OR) values of MS, logistic regression analyzes were performed to examine the relationship between those obesity indices and MS after correcting for identified or chosen confounders in various models. There was no covariate adjustment in Model 1. Age, education level, smoking, alcohol consumption, and exercise were all modified in Model 2. The area under the ROC curve (AUC) was used to calculate the receiver operating characteristic curve (ROC), which was used to assess the contribution of the factors to the identification of discriminatory power for MS (MedCalc 19.8). The highest point of the Youden index was selected as the optimal cutoff point for the obesity index for the prediction of MS. Accuracy, sensitivity, specificity, Youden index, positive predictive value (PPV), and positive likelihood ratio (+LR) were the evaluation measurements selected for this study. The level of statistical significance was established at a threshold of *p* < 0.05.

## 3. Results

The demographic, anthropometric measurements and clinical characteristics of subjects are shown in Table 1. The results of the Kolmogorov-Smirnov test indicated that all continuous variables in this study did not conform to a normal distribution, therefore, continuous-type variables were described by a median and interquartile range (IQR) and analyzed by the Kruskal-Wallis H test. There were 14,391 males (40.6%) and 21,055 (59.4%) females. A total of 60.8% of men had six or more years of education, compared with only 48.1% of women. Meanwhile, compared with women, men had significantly higher smoking rates (57.5% vs. 0.3%), alcohol drinking rates (32.0% vs. 0.8%), and the observed differences exhibited statistical significance (*p* < 0.001). With regard to the anthropometric measurement indicator, men had a higher BMI, waist circumference (WC), waist-hip ratio (WHR), and waist-height ratio (WHtR), but a lower LAP, and the observed differences reached statistical significance. Blood lipids, HDL cholesterol, LDL cholesterol, and total cholesterol in women were significantly higher than those in men, but triglycerides in men were significantly higher, and the difference was statistically significant (Table 1).

The overall prevalence of MS was 31.7%, and the prevalence exhibited a statistically significant gender difference, with lower rates observed in males compared to females (27.0% vs. 34.9%, *p* < 0.001). The LAP of the population was 27.9 (15.9–46.5) cm·mmol/L, 27.20 (15.18–46.00) cm·mmol/L for males and 28.48 (16.35–46.72) cm·mmol/L for females. The prevalence of MS increased with increasing LAP, and the trend was the same for both men and women (*p* < 0.001, Table 1, Figure 1).

Univariate logistic regression analysis showed that LAP, BMI, WC, WHR, and WHtR were associated with MS (*p* < 0.001). After adjusting for all covariates (model 2), the multiple logistic regression analyzes showed that all the indices, including LAP(OR_male_ = 1.074, OR_female_ = 1.097), BMI(OR_male_ = 1.463, OR_female_ = 1.334), WC(OR_male_ = 1.178, OR_female_ = 1.134), WHR(OR_male_ = 1.188, OR_female_ = 1.152), and WHtR(OR_male_ = 1.292, OR_female_ = 1.202) were associated with MS (Table 2).

The ROC curve and related evaluation index data were shown in Figure 2 and Table 3. The cutoff point of the LAP for men and women was 36.04 (Se: 81.91% and Sp: 81.06%) and 34.95 (Se: 80.93% and Sp: 83.04%). In terms of accuracy, the proportion of samples correctly predicted by the LAP was higher than that by the other measures, and this advantage was particularly significant in women. The corresponding Youden index of LAP was also the highest among all indicators, with a value of 0.64 for both sexes. It was found that the positive predictive values obtained by LAP (67.24%) were significantly higher than other obesity indicators (*p* < 0.01), both in men and women. The LAP had a positive likelihood ratio of 4.454 and 4.77 for men and women, respectively (Figure 2, Table 3).

## 4. Discussion

This study presents a comprehensive account of the relationship between LAP and MS in the Songjiang District, Shanghai, China. To our knowledge, this is the first population-based investigation to assess the prognostic value of LAP for identifying MS among the general populace in China. In this research, a cross-sectional survey of 360,403 community residents aged 20–75 years in the Songjiang District, Shanghai, China was used to analyze the results. It was discovered that LAP enhanced the risk of MS in both men and women and that the area under the LAP curve was much higher than other obesity indicators. In addition, LAP had a relatively high sensitivity and specificity, as well as a relatively high positive likelihood ratio and positive predictive value. This demonstrates that LAP may be used as a screening metric for the general population in mainland China.

LAP is a novel indicator of obesity that accounts for visceral fat accumulation. It combines WC, which measures abdominal adipose tissue, with TG, which indicates the degree of visceral fat accumulation brought on by metabolic problems. Obesity is currently a significant global health issue for all nations. Asians have a lower BMI than whites, but they tend to have visceral fat deposits [31]. Therefore, LAP is a better indicator of obesity in Asians than BMI. In addition, there are many diagnostic criteria for MS, but no matter which one is chosen, the information on five aspects of the subject needs to be collected, and some aspects have more than one index, which is time-consuming and not conducive to screening. However, LAP, if chosen as a judging criterion for screening MS, is simpler and more direct, and can be quickly calculated to make a judgment.

In our research, the total prevalence of MS was 31.7%. However, there was a significant difference from previous studies [32,33,34,35,36], which suggests that the indicators and thresholds for the judgment and screening of MS may vary from place to place [1]. At the same time, people’s diets and living habits are also changing dramatically, and the prevalence of MS in the same region may also change dramatically [37]. Therefore, it is necessary and urgent to conduct research on MS predictive indicators and their critical values in order to anticipate early detection of the disease and prevent its development.

MS is a disease closely related to obesity [31,38], therefore, it is reasonable and appropriate to consider the use of obesity indicators to predict MS. Currently, common indicators to evaluate obesity include BMI, WC, WHR, and WHtR. These obesity assessment metrics are easy to measure but also have some deficiencies in practical applications. For example, BMI primarily assesses general obesity but does not differentiate between the percentage of muscle and fat in a person’s body. Waist circumference indicators (WC, WHR, WHtR) can indicate abdominal obesity but they are unable to differentiate between visceral and subcutaneous fat [31,39]. Furthermore, research has indicated that although Asians tend to have a lower body mass index than Whites, they are more likely to have visceral fat depositions [31]. In short, it can be seen that these indicators are somewhat deficient in measuring body fat content, especially visceral fat. The LAP, a new indicator to measure obesity, is a combination of WC, which measures abdominal fat, and TG, which reflects dyslipidemia, and is a good proxy for the extent of visceral fat accumulation due to metabolic disorders.

The findings of this research demonstrated a correlation between MS and all obesity indicators across all genders, including LAP, BMI, WC, WHR, and WHtR, which was consistent with the findings in related research among the undiagnosed Brazilian adults [40], indicating that visceral, systemic, and abdominal obesity are all intimately associated with the occurrence and progression of MS. With an overall AUC value of 0.898 (95%CI: 0.894–0.901), LAP was found to have the strongest predictive power in our study. The outcomes imply that LAP can be used to predict MS in the general population of China. The LAP demonstrated an advantage in predicting metabolic syndrome in different populations, with an AUC value larger than 0.85. For example, the AUC for metabolic syndrome in the Indian population was 0.901 (95%CI: 0.85–0.95) [41]; the AUC of Chinese adults and seniors with diabetes was 0.887 (95%CI: 0.852–0.922) [42]; the LAP also demonstrated the greatest accuracy in predicting MS among obesity measures with an AUC of 0.901 (95%CI: 0.870–0.932) in Taiwanese citizens aged 50 and over [39]; the AUC for MS in healthy men from Buenos Aires was 0.91 [43]. Although different definitions of metabolic syndrome were used in a 2016 research study in northern Iran, Amol [2] and in a 2017 survey of undiagnosed Brazilian adults [40], we could see that no matter which definition was used, the AUC obtained by the LAP in the diagnosis of MS was always higher than other obesity indicators.

The cut-off values in previous studies are somewhat different from ours [2,40], which may be attributed to many factors such as economic and social development, regional, lifestyle, dietary habits, race, the selection of the study population, sample size, MS diagnostic criteria, and so on. However, the LAP fared better than other indicators in terms of sensitivity and specificity, as well as the positive predictive value and likelihood ratio, showing that LAP has a significant advantage as a predictor of MS in the Chinese general population.

There are three main aspects to our innovation. First of all, the conventional diagnosis of metabolic syndrome often needs to check five indicators and determine whether the patient meets three or more of the conditions, which are costly, time-consuming, and laborious. However, our study attempts to use only one obesity-related indicator to judge MS, simplifying the process and saving manpower, money, material resources, and time. Secondly, our study pioneered the use of a novel indicator, the lipid accumulation index, to diagnose metabolic syndrome in the Chinese population and to comprehend the practicality of this indicator in the broader Chinese population; last but not least, our study had an ample sample size, a strict quality control system, and it recruited participants from a community-based population. The findings are trustworthy and convincing and may be used to gain insight into the current situation in China. At the same time, there are a few limitations that should be mentioned. Firstly, a cross-sectional study design served as the foundation for our analyzes, so there are some restrictions on extrapolating the study’s findings. Secondly, there are still some confounding factors that have not been considered and will have a certain impact on the study conclusions. Therefore, future prospective studies are required to look into the accuracy of the LAP for MS diagnosis.

## 5. Conclusions

The LAP was discovered to be a reliable and straightforward tool for diagnosing MS in the contemporary Chinese general population, and it could be utilized by physicians efficiently, particularly in screening. This practical tool may be helpful in a primary care context to pinpoint patients who need more testing and care.

## Figures and Tables

**Figure 1 nutrients-15-03168-f001:**
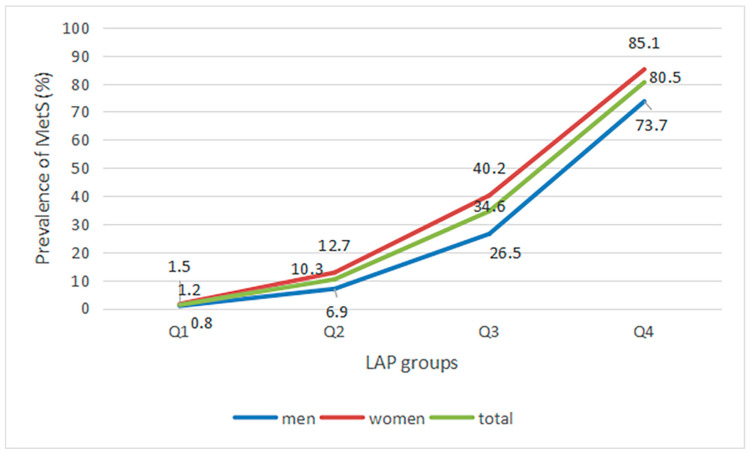
Prevalence of MS in different LAP groups.

**Figure 2 nutrients-15-03168-f002:**
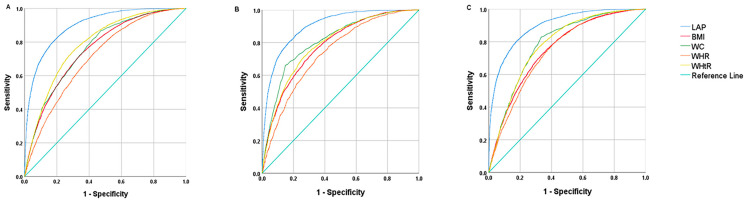
The ROC curves of various indicators for the prediction of MS for all population (**A**), men (**B**) and women (**C**).

**Table 1 nutrients-15-03168-t001:** The demographic, anthropometric measurements and clinical characteristics of participants.

Variable	Male (*n* = 14,391)	Female (*n* = 21,055)	Total	*p* Value
Age (years)				
20-	1903 (13.2)	3147 (14.9)	5050 (14.2)	<0.001
45-	3304 (23.0)	5943 (28.2)	9247 (26.1)	
55-	4802 (33.4)	6992 (33.2)	11,794 (33.3)	
65-	4382 (30.4)	4973 (23.6)	9355 (26.4)	
Education level (years)				
<6	5633 (39.1)	10,930 (51.9)	16,563 (46.7)	<0.001
6-	7878 (54.7)	8848 (42.0)	16,726 (47.2)	
10-	880 (6.1)	1277 (6.1)	2157 (6.1)	
Smoking				
Yes	8276 (57.5)	63 (0.3)	8339 (23.5)	<0.001
No	6115 (42.5)	20992 (99.7)	27,107 (76.5)	
Alcohol drinking				
Yes	4607 (32.0)	159 (0.8)	4766 (13.4)	<0.001
No	9784 (68.0)	20,896 (99.2)	30,680 (86.6)	
Exercise				
Yes	4615 (32.1)	6663 (31.7)	11278 (31.9)	0.407
No	9743 (67.9)	14,340 (68.3)	24,083 (68.1)	
BMI (kg/m^2^)				
<18.5	269 (1.9)	653 (3.1)	922 (2.6)	<0.001
18.5-	5718 (39.7)	10,064 (47.8)	15,782 (44.5)	
24-	6339 (44.1)	7624 (36.2)	13,963 (39.4)	
28-	2063 (14.3)	2711 (12.9)	4774 (13.5)	
LAP (cm·mmol/L)	27.2 (15.2–46.0)	28.5 (16.4–46.7)	27.9 (15.9–46.5)	<0.001
Waist circumference (cm)	85.0 (79.0–90.3)	79.6 (73.0–86.0)	82.0 (75.0–88.0)	<0.001
WHR (%)	90.2 (86.6–93.6)	86.8 (82.3–90.9)	88.3 (83.9–92.3)	<0.001
WHtR (%)	51.1 (47.5–54.4)	51.0 (46.8–55.4)	51.0 (47.1–55.0)	0.016
Systolic BP (mm Hg)	132.0 (121.0–144.0)	132.0 (119.3–146.0)	132.0 (120.0–146.0)	0.002
Diastolic BP (mm Hg)	81.0 (75.3–88.0)	78.0 (72.0–85.7)	80.0 (73.0–86.0)	<0.001
Fasting glucose (mmol/L)	4.7 (4.3–5.4)	4.7 (4.3–5.4)	4.7 (4.3–5.4)	0.53
HbA1C (mmol/mol)	5.6 (5.3–6.0)	5.6 (5.3–6.0)	5.6 (5.3–6.0)	0.95
Triglycerides (mmol/L)	1.4 (1.0–2.0)	1.3 (1.0–1.9)	1.4 (1.0–1.9)	<0.001
HDL cholesterol (mmol/L)	1.3 (1.1–1.4)	1.5 (1.3–1.7)	1.4 (1.2–1.6)	<0.001
LDL cholesterol (mmol/L)	2.7 (2.2–3.2)	2.8 (2.3–3.3)	2.7 (2.2–3.3)	<0.001
Total cholesterol (mmol/L)	4.7 (4.2–5.3)	5.0 (4.4–5.6)	4.9 (4.3–5.5)	<0.001

The median (interquartile range) or *n* (%) of the data is displayed.

**Table 2 nutrients-15-03168-t002:** Logistic regression models evaluating the associations of obesity indicators with MS.

	Male OR(95%CI)		Female OR(95%CI)	
	Model 1	Model 2	Model 1	Model 2
LAP	1.071 (1.068–1.073) *	1.074 (1.071–1.077) *	1.101 (1.098–1.104) *	1.097 (1.094–1.100) *
BMI	1.462 (1.438–1.485) *	1.463 (1.439–1.486) *	1.360 (1.345–1.375) *	1.334 (1.319–1.349) *
WC	1.178 (1.170–1.185) *	1.178 (1.171–1.186) *	1.145 (1.140–1.150) *	1.134 (1.129–1.139) *
WHR	1.184 (1.173–1.196) *	1.188 (1.177–1.199) *	1.167 (1.159–1.174) *	1.152 (1.144–1.159) *
WHtR	1.274 (1.262–1.287) *	1.292 (1.278–1.305) *	1.212 (1.205–1.220) *	1.202 (1.194–1.211) *

Model 1: an unadjusted model; Model 2: adjusted for age, education level, smoking, alcohol drinking, exercise. * *p* < 0.001.

**Table 3 nutrients-15-03168-t003:** The ROC analysis of various obesity indicators for MS prediction by sex.

Gender	Variable	Cut-Off Point *	Accuracy	Sensitivity (%)	Specificity (%)	Youden Index	PPV (%)	+LR
Male	LAP	>36.25	0.80	81.91	81.6	0.64	62.22	4.45
	BMI	>25.09	0.68	74.94	67.73	0.43	46.21	2.32
	WC	>89.67	0.79	67.25	85.15	0.52	62.63	4.53
	WHR	>90.86	0.64	70.95	64.82	0.36	43.2	2.01
	WHtR	>52.29	0.76	73.91	71.76	0.46	49.15	2.61
Female	LAP	>34.95	0.81	80.93	83.04	0.64	71.87	4.77
	BMI	>23.97	0.67	75.56	64.63	0.40	53.34	2.13
	WC	>79.90	0.71	82.98	68.29	0.51	58.36	2.62
	WHR	>85.99	0.54	78.11	59.57	0.38	54.11	1.93
	WHtR	>51.49	0.68	77.9	68.85	0.47	57.26	2.5

* LAP is measured in cm*mmol/L; BMI is measured in kg/m^2^; WC is measured in cm; Abbreviation: PPV: Positive Predictive Value; +LR: Positive likelihood ratio.

## Data Availability

Upon reasonable request, the corresponding author will provide the dataset that was utilized and examined during the current investigation.

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
