# Peer review of "Enhanced Predictive Value of Lipid Accumulation Product for Identifying Metabolic Syndrome in the General Population of China"

_nutrients, 2023, doi:10.3390/nu15143168_

Round 1
Reviewer 1 Report
I would like to congratulate the authors for having carried out a study in the area of obesity and metabolic syndrome, which, as they well comment, is beginning to be a problem even in populations such as China, where until recently they had a low prevalence.
The paper has been carried out in a large population, which is a good point, but I have a great concern about the novelty of this work itself. The index is not new, so various previous studies have shown its usefulness following a similar methodology. In fact, as the authors themselves comment, the use of the LAP as a predictor of metabolic syndrome has been used in other population groups, obtaining similar data, so the only novelty of the study is that it was done in the Chinese population. At this point, the usefulness of such research is questionable, since I do not believe it is necessary to replicate the same experiment in all populations.
Nevertheless, the objectives are clear, the methodology is adequate, and the writing of the document is very good.
I only have a few minor comments:
The abstract is clear and well structured
Introduction:
I miss a better or more extended definition of metabolic syndrome, since the paper is focused on this disease.
Line 49, kahn should be in capital character.
Methods:
More details are needed for statistical analysis. why did the authors choose non-parametric tests? Attending to the high sample size, I consider inadequate to use such tests.
I will define the Youden index as well as the other indicators of prediction (sensitivity, sensibility, PPV, +LR, etc.)
More details are neede for the logistic procedure. For instance, all the parameters included have a great colinearity. How was this controlled? Why the metabolic parameters were not included?
Results:
In table 1, there are some differences that don't seem to be as significant as described, for instance for LDL cholesterol. How do you adjust for multiple comparisons?
Table 3. You can include more prediction parameters such as accuracy, etc...
Author Response
请参阅附件。

Reviewer 2 Report
The authors examine the diagnostic capacity of LAP for MS in comparison to several established indicators of MS.
The study is well designed and data logically presened and discusses.
The strength of the study is a huge sample size, however, my major concern is the lack of novelty.It is novel for Chinese population but as the authors acknowledged discussiing papers describing LAP as a diagnostic tool for MS in other populations, the role of LAP for identifying MS has already been used in the past in other populations.
Minors:
Methods: subchaper 2.2.3.
Were plasma or serum used for the analyses? It says the specimens were stored at -80°C ; plasma or serum?
Was HbA1c measured in fresh blood or after freezing and thawing?
Lanes 233-236: The sentence is too long and not clear. Please improve.
Engslish is fine, some sentences shoud be shorter and more clear.
